# Cytosolic sequestration of the vitamin D receptor as a therapeutic option for vitamin D-induced hypercalcemia

Daniela Rovito [1,2,3,4], Anna Y. Belorusova [5], Sandra Chalhoub[1,2,3,4], Anna-Isavella Rerra[1,2,3,4], Elvire Guiot[1,2,3,4], Arnaud Molin[6,7], Agnès Linglart[7,8], Natacha Rochel [1,2,3,4], Gilles Laverny [1,2,3,4✉] & Daniel Metzger [1,2,3,4✉]

The bioactive vitamin $D_3$, $1\alpha,25(OH)_2D_3$, plays a central role in calcium homeostasis by controlling the activity of the vitamin D receptor (VDR) in various tissues. Hypercalcemia secondary to high circulating levels of vitamin $D_3$ leads to hypercalciuria, nephrocalcinosis and renal dysfunctions. Current therapeutic strategies aim at limiting calcium intake, absorption and resorption, or $1\alpha,25(OH)_2D_3$ synthesis, but are poorly efficient. In this study, we identify WBP4 as a new VDR interactant, and demonstrate that it controls VDR sub-cellular localization. Moreover, we show that the vitamin D analogue ZK168281 enhances the interaction between VDR and WBP4 in the cytosol, and normalizes the expression of VDR target genes and serum calcium levels in $1\alpha,25(OH)_2D_3$-intoxicated mice. As ZK168281 also blunts $1\alpha,25(OH)_2D_3$-induced VDR signaling in fibroblasts of a patient with impaired vitamin D degradation, this VDR antagonist represents a promising therapeutic option for $1\alpha,25(OH)_2D_3$-induced hypercalcemia.

[1] Institut de Génétique et de Biologie Moléculaire et Cellulaire, Illkirch, France. [2] Centre National de la Recherche Scientifique, UMR7104 Illkirch, France. [3] Institut National de la Santé et de la Recherche Médicale (INSERM), U1258 Illkirch, France. [4] Université de Strasbourg, Illkirch, France. [5] Medicinal Chemistry, Respiratory, Inflammation and Autoimmunity, BioPharmaceuticals R&D, AstraZeneca, Gothenburg, Sweden. [6] Université de Normandie, UNICAEN, CHU de Caen Normandie, Service de Génétique, EA 7450 BIOTARGEN Caen, France. [7] Reference Center for Rare Diseases of Calcium and Phosphorus Metabolism (OSCAR), Paris, France. [8] Université de Paris Saclay, AP-HP, Hôpital Bicêtre, DMU SEA, INSERM, U1185, Le Kremlin Bicêtre, France. ✉email: laverny@igbmc.fr; metzger@igbmc.fr

Calcium homeostasis is controlled by a network involving the parathyroid glands (PTG), intestine, kidney, and bones. At low serum calcium levels, the PTG secretes the parathyroid hormone (PTH), which enhances the hydroxylation of 25(OH)vitamin $D_3$ by CYP27B1 into its bioactive form $1\alpha,25$ $(OH)_2$vitamin $D_3$ (1,25D3) in the kidney. 1,25D3 induces the transcriptional activity of the vitamin D receptor (VDR, also known as NR1I1)[1,2], and thereby increases calcium absorption in intestine, calcium reabsorption in kidney, and calcium resorption in bones[3]. Importantly, high 1,25D3 levels (hypervitaminosis D) lead to hypercalcemia, and shut down PTH production in the PTG via calcium-dependent but VDR-independent pathways[4].

Hypervitaminosis D in patients with granulomatous diseases, lymphomas, or vitamin D 24-hydroxylase (CYP24A1) loss-of-function mutations (also known as idiopathic infantile hypercalcemia (IIH)[5]) leads to hypercalciuria, nephrocalcinosis, and renal dysfunctions[5–8]. Hypercalcemia in such patients can be reduced by diuretics in combination with limiting calcium and vitamin D intake, but in severe cases, treatments based on bisphosphonates, ketoconazole (KTZ), or glucocorticoids are required to decrease bone calcium resorption and/or intestinal calcium absorption. However, these treatments do not have sustained effects, have major drawbacks, and dramatically impact the development of children and the quality of life of patients[6].

More than 3000 1,25D3 analogs were synthetized by medicinal chemistry approaches to modulate VDR activities, including many agonists and few antagonists[9]. Among the latter, ZK168281 (ZK), a carboxylic ester analog of 1,25D3 with a rigid extended side chain and an ethyl acrylate moiety, was shown to bind to VDR with a similar affinity than 1,25D3[10], and to reduce 1,25D3-induced gene expression in human and rat cell culture systems[11–13]. Moreover, crystal structure and hydrogen deuterium exchange (HDX) analyses of zebrafish VDR ligand binding domain (LBD) revealed that ZK does not induce the active conformation of VDR[14], in agreement with previous molecular dynamic studies[15]. However, ZK-mediated VDR antagonistic activities in vivo, and the underlying molecular mechanisms remained to be determined.

In this work we show that ZK is a VDR antagonist in rat intestinal cells and in human fibroblasts, whereas CYP24A1-generated ZK derivatives have partial agonistic activities. Importantly, ZK blunts 1,25D3-induced gene expression by enhancing the interaction of VDR with WBP4 (also known as FBP21) in the cytosol, thereby impairing its nuclear translocation. Moreover, ZK normalizes serum calcium levels in 1,25D3-intoxicated mice, and 1,25D3-induced VDR signaling in fibroblasts of an IIH patient. Thus, this compound represents a potent and safe vitamin D analog for the treatment of 1,25D3-induced hypercalcemia.

## Results

### ZK exerts VDR agonistic and antagonistic activities.
To further characterize ZK activities, IEC-18 rat intestinal epithelial cells were treated for 4 h with vehicle, 100 nM 1,25D3, 1 μM ZK, or a combination of both ligands. The transcript levels of the VDR target gene *Cyp24a1*, encoding the main 1,25D3 catabolic enzyme[16], as well as those involved in calcium absorption (i.e. *S100g*, *Atp2b1* and *Trpv6*)[2] were increased by at least 3-fold by 1,25D3 and by ZK. In contrast, they were not induced by a 1,25D3 and ZK co-treatment (Fig. 1a), indicating that ZK exerts VDR agonistic and antagonistic activities.

While *Cyp24a1* transcript levels were induced by 3-, 9-, and 16-fold after a 2, 4, and 6 h 1,25D3 treatment, respectively, no induction was observed after a co-treatment with 1,25D3 and ZK at any time point (Fig. 1b). VDR protein levels were increased by

1.2- to 1.7-fold after 2–6 h treatments of 1,25D3 and ZK, alone or in combination (Fig. 1c, d), demonstrating that ZK antagonistic activities do not result from VDR degradation. Even though VDR protein levels were higher in cells treated for 2 h with ZK than with 1,25D3 (Fig. 1c, d), ZK treatments induced *Cyp24a1* transcripts with a 2 h delay compared to 1,25D3 (Fig. 1b), indicating that ZK metabolites might exert VDR agonistic activities. As cytochrome P450 enzymes not only induce 1,25D3 catabolism, but also enhance the activity of some vitamin D analogs[16], the effects of ZK were determined in IEC-18 cells pre-treated with KTZ, a broad-spectrum inhibitor of such enzymes[17]. Under these conditions, ZK-induced VDR target gene expression was abolished, while its antagonistic activities on 1,25D3-induced genes were not affected (Fig. 1a), indicating that ZK derivatives produced in cells, but not the parent compound, have VDR agonistic activities. To investigate whether ZK modifications are CYP24A1 dependent, ZK activities were determined in fibroblasts from an adult patient with hypercalcemia carrying CYP24A1 loss-of-function mutations[18] (FB-CYP) and in the human fibroblastic cell line FB-789[19], used as control. CYP24A1 protein was detected in FB-789, but not in FB-CYP cells (Fig. 1e), demonstrating that the mutations induce protein loss. ZK prevented 1,25D3-induced *CYP24A1* transcripts in both cell types, but induced *CYP24A1* transcripts only in FB-789 cells (Fig. 1f). Thus, ZK is a potent VDR antagonist, and its CYP24A1-generated metabolite(s) exert agonistic activities.

### ZK impairs 1,25D3-induced VDR nuclear translocation.
VDR regulates the expression of target genes as a heterodimer with retinoid X receptors (RXRs)[20], and RXR was shown to be required for ZK-induced VDR antagonism[11]. To characterize conformational changes of VDR/RXR heterodimers upon ZK binding, we performed HDX coupled with mass spectrometry (MS) analyses using full-length hVDR and N-terminal domain truncated hRXRα (hRXRαΔNTD) [Supplementary Table 1 and Source Data]. No detectable perturbation was observed in the DNA binding domain (DBD) of hRXRα (Supplementary Fig. 1a) and of hVDR (Fig. 2a and Supplementary Fig. 1b) upon binding of either 1,25D3 or ZK. Moreover, deuterium exchange rates of the dimerization interface that involves residues from helices (H) 7, H9, and H10 and of the loop between H8 and H9 of both proteins[21,22] were similar upon 1,25D3 and ZK binding (Fig. 2a, b and Supplementary Fig. 1a–f). In addition, surface plasmon resonance (SPR) revealed that the affinity of the LBD of hVDR (aa 118–427) for that of hRXRα (aa 223–462) increased to a similar extent upon ZK and 1,25D3 binding (Supplementary Fig. 1g, h). Thus, 1,25D3- and ZK-induced hVDR conformations promote hVDR/hRXRαΔNTD heterodimer formation.

In agreement with previous results[23], 1,25D3 binding protected hVDR amino acids 224-233, 263-269, 300-307, 390-401, and 414-419 located in H3, H5, H6-H7, H10-H11, and H12, respectively (Fig. 2a, c). In contrast, upon ZK binding, deuterium exchange of H3, H5, and H6 was 6–11% lower than upon 1,25D3 binding, and that of hVDR H12 was similar to that of unliganded receptor (Fig. 2a–c and Supplementary Fig. 1b). As 1,25D3-induced hVDR H12 folding is mandatory for VDR activity, these results show that ZK does not induce the hVDR active conformation, in agreement with molecular dynamic simulations[15].

Importantly, the higher sequence coverage than in the previously reported HDX-MS analysis of zebrafish VDR LBD[14] revealed that the binding of 1,25D3 protects hVDR residues 139-158, encompassing a nuclear localization signal (NLS)[24], from deuterium exchange (Fig. 2a, d). Deuterium exchange of the NLS was decreased by about 45% by 1,25D3 and by ZK at early time points. In contrast, after 10 min, it was reduced only by 1,25D3

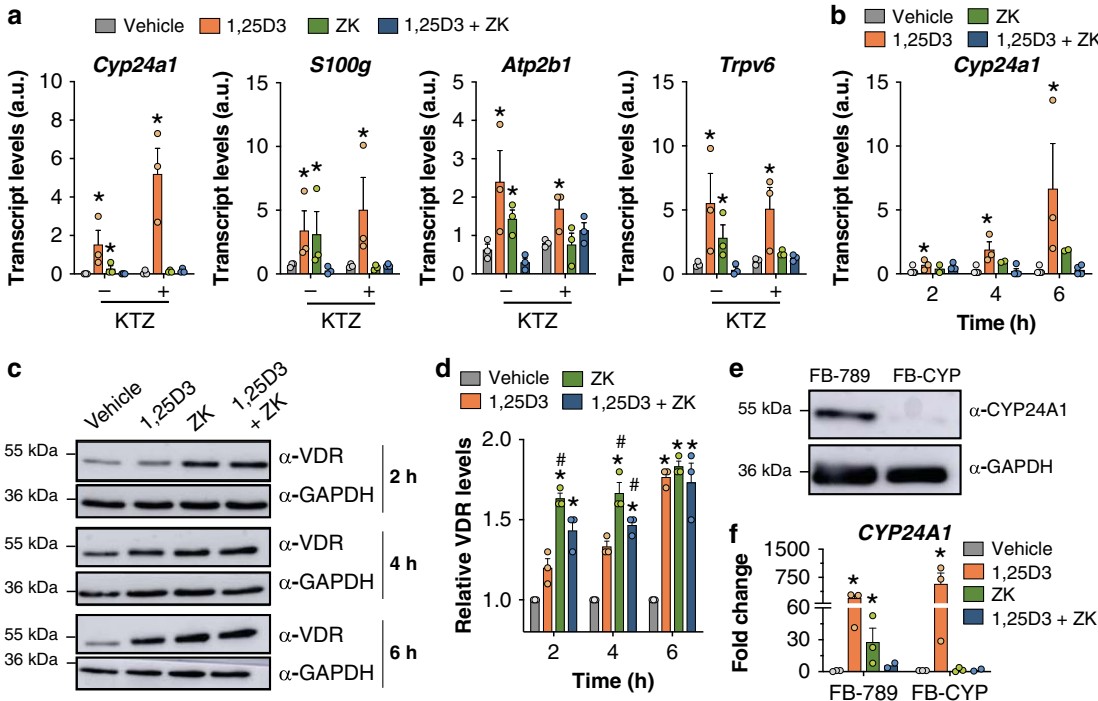

**Fig. 1 ZK activities in rat intestinal epithelial cells and in human fibroblasts. a** Relative transcript levels of *Cyp24a1*, *S100g*, *Atp2b1*, and *Trpv6* after a 4 h treatment with vehicle, 100 nM 1,25D3, 1 μM ZK, or 100 nM 1,25D3 and 1 μM ZK in IEC-18 cells, pre-treated (+) or not (−) with 5 μM ketoconazole (KTZ) for 2 h. $n = 3$ independent biological replicates/condition. **b** Relative *Cyp24a1* transcript levels in IEC-18 cells treated with vehicle, 100 nM 1,25D3, 1 μM ZK, or 100 nM 1,25D3 and 1 μM ZK for 2, 4, and 6 h. $n = 3$ independent biological replicates for vehicle, 1,25D3, and 1,25D3 and ZK. $n = 2$ independent biological replicates for ZK. Representative VDR immunoblot (**c**) and average fold-change (**d**) of total extracts of IEC-18 cells treated for 2, 4, and 6 h with vehicle, 100 nM 1,25D3, 1 μM ZK, or 100 nM 1,25D3 and 1 μM ZK. $n = 3$ independent biological replicates/condition. Unprocessed blots in Source Data. **e** Immunoblot of CYP24A1 protein in human FB-789 and FB-CYP fibroblasts. GAPDH was used as loading control. Unprocessed blots in Source Data. **f** Fold change of *CYP24A1* transcript levels in FB-789 and in FB-CYP fibroblasts treated for 6 h with vehicle, 100 nM 1,25D3, 1 μM ZK, or 100 nM 1,25D3 and 1 μM ZK. The levels of vehicle-treated cells were set as reference. $n = 3$ independent biological replicates/condition. Data in (**a**), (**b**), (**d**), and (**f**) are represented as mean + s.e.m. (standard error of mean). *$p < 0.05$ vs. vehicle, #$p < 0.05$ vs. 1,25D3; one-way ANOVA with Tukey post-hoc test. The exact significant *p*-values are provided in Supplementary Table 4.

(Fig. 2a, d, e and Supplementary Fig. 1b), indicating that ZK less efficiently stabilizes the NLS conformation.

To determine whether ZK impairs VDR intracellular trafficking, cytosolic and nuclear extracts of FB-789 cells treated for 1.5 h with vehicle or 10 nM 1,25D3, or co-treated with 10 nM 1,25D3 and 100 nM ZK were analyzed. Cytosolic VDR levels were 2-fold lower in 1,25D3-treated than in 1,25D3 and ZK co-treated cells. In contrast, in the nuclear fraction, VDR levels were 6-fold higher in 1,25D3-treated cells than in vehicle, and a ZK co-treatment reduced them by 3-fold (Fig. 3a, b). In agreement with these results, a ZK co-treatment reduced 1,25D3-induced VDR nuclear localization by 2-fold in IEC-18 cells (Fig. 3c, d). In contrast, VDR was predominantly in the nuclear fraction in the presence of ZK (Fig. 3c, d). Moreover, KTZ did not affect VDR localization in 1,25D3-treated, and in 1,25D3 and ZK co-treated cells, but impaired ZK-induced VDR nuclear localization (Fig. 3e, f), showing that ZK intracellular derivatives promote VDR nuclear translocation, and that ZK antagonizes 1,25D3 activities by sequestrating VDR in the cytosol.

**ZK enhances the VDR/WBP4 interaction in the cytosol.** To identify ZK-liganded VDR interactants that might affect VDR intracellular localization, proteins co-immunoprecipitated with cytosolic VDR from IEC-18 cells were analyzed by MS. As expected, cellular component annotations showed that most of the immunoprecipitated proteins enriched in 1,25D3, and ZK co-treated cells have a cytosolic localization (Supplementary

Table 2). Surprisingly, peptides of WBP4, a protein previously reported as a component of the spliceosome[25,26], were highly enriched in ZK and 1,25D3 co-treated cells compared to vehicle-treated cells (Fig. 4a), and were not detected in 1,25D3-treated cells (Fig. 4b). Note that none of the previously identified WBP4 nuclear partners (e.g. BRR2 and PRP8)[27] were detected (full list provided as Source Data file Fig. 4a). The interaction between WBP4 and VDR was confirmed by VDR immunoblotting in WBP4-immunoprecipitated cytosolic extracts of IEC-18 and of FB-789 cells co-treated with 1,25D3 and ZK (Fig. 4c and Supplementary Fig. 2a, b). Of note, only a weak interaction between WBP4 and VDR was observed in basal conditions (Fig. 4c and Supplementary Fig. 2b).

WBP4 contains two WW domains (aa 122–196 in human) that are docking sites for proline-rich motifs[27]. As ZK destabilized VDR amino acids 152–158 encompassing a diproline motif (Fig. 2d–e), VDR might directly interact with WBP4. Native gel experiments coupled to MS analyses revealed that the WBP4 polypeptide encompassing the WW domains is bound to VDR in the presence of ZK (Supplementary Fig. 2c). Moreover, micro-scale thermophoresis confirmed this direct interaction, and indicated that the affinity between the two polypeptides is increased by about 2-fold in the presence of ZK (Supplementary Fig. 2d). Thus, VDR interacts with WBP4 through the two WW domains and ZK enhances this interaction.

Immunostaining and immunoblotting revealed that endogenous WBP4 was both nuclear and cytosolic in IEC-18 cells treated with vehicle or 1,25D3, and in cells co-treated with

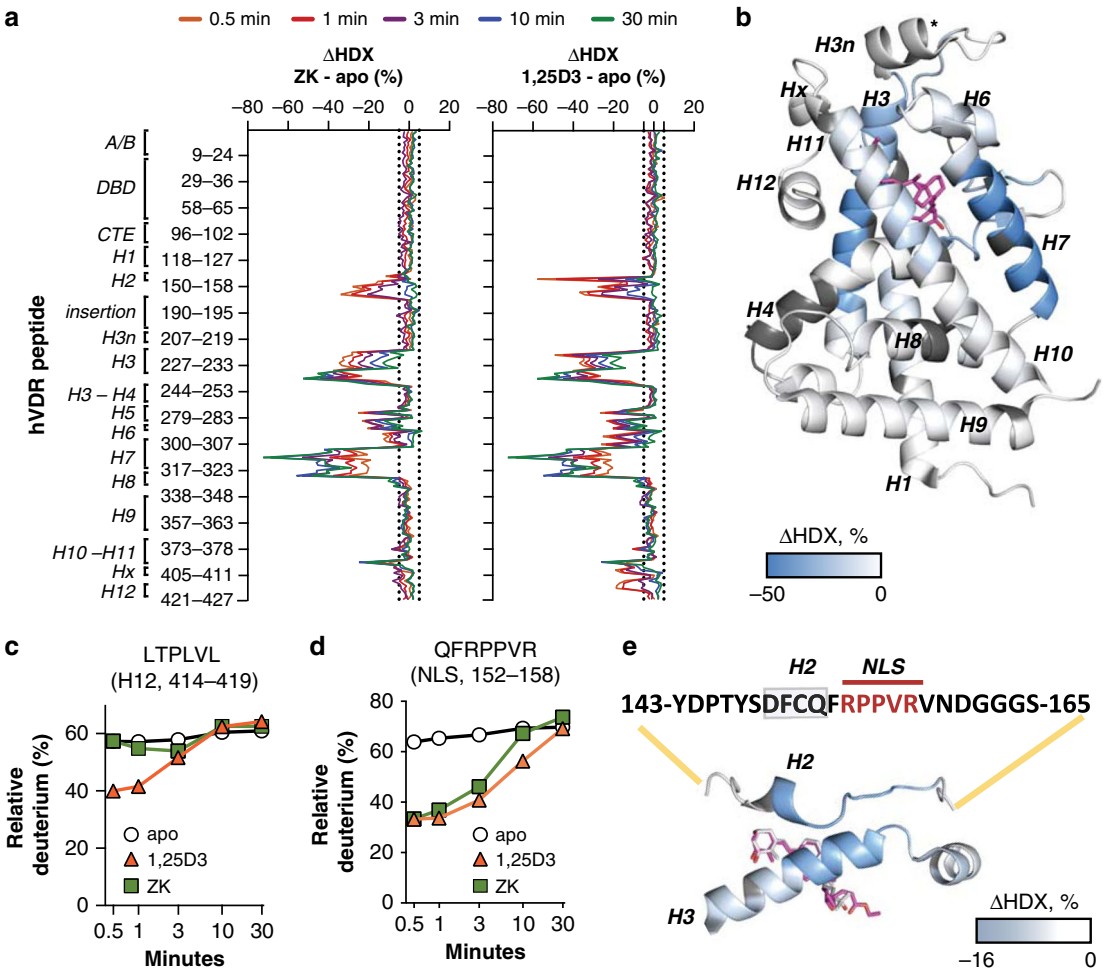

**Fig. 2 Ligand-induced hVDR conformational changes. a** Differential deuterium uptake (ΔHDX) of hVDR between apo and ZK-bound (left panel) or 1,25D3-bound (right panel) hVDR/hRXRαΔNTD heterodimers. The VDR LBD is composed of H1–H12. Negative values indicate decreased deuterium incorporation upon ligand binding. Dashed lines represent a ΔHDX of 5%. **b** Average ΔHDX of hVDR between ZK-bound and apo-hVDR/hRXRαΔNTD heterodimers across the various time points mapped onto hVDR LBD (PDB: 1DB1[44]). Regions not covered by HDX-MS are shown in dark gray. Differential deuterium uptake plots of the hVDR residues 414-LTPLVL-419 located in H12 (**c**) and of 152-QFRPPVR-158 located in the H1–H3 loop (**d**) determined in ZK-bound, 1,25D3-bound and apo hVDR/hRXRαΔNTD heterodimers. **e** Zoom of average ΔHDX across the various time points of the VDR region encompassing H2 and a nuclear localization signal (NLS) mapped onto the hVDR LBD (PDB: 1DB1[44]). HDX-MS data were collected in triplicates, and deuteration data were normalized to the maximal theoretical uptake (±s.e.m.).

1,25D3 and ZK (Fig. 4d and Supplementary Fig. 2e). Moreover, two molecular weight species at about 55 kDa were detected by immunoblotting nuclear and cytosolic fractions of IEC-18, U2OS, FB-789, and HeLa cells with WBP4 antibodies (Supplementary Fig. 2e), in agreement with the manufacturer's datasheet. Note that these two protein species were differentially expressed in cytosolic and nuclear fractions. Small interfering RNA (siRNA)-mediated silencing of WBP4 in IEC-18 cells reduced its transcript and protein levels by more than 80% (Supplementary Fig. 3a, b), and markedly decreased both cytosolic and nuclear WBP4 immunostaining (Supplementary Fig. 3c, d). Importantly, VDR was mainly nuclear in WBP4-silenced cells (Fig. 4e and Supplementary Fig. 3e, f), and the transcript levels of *Cyp24a1*, *S100g* and *Atp2b1* were at least 3-fold higher than in control cells (Fig. 4f). Moreover, WBP4 silencing had almost no effect on 1,25D3-induced VDR target gene expression, but impaired ZK antagonistic activities (Fig. 4g). Thus, these results demonstrate that WBP4 modulates the distribution of VDR between the cytosol and the nucleus, and that ZK-enhanced VDR/WBP4 interaction in the cytosol underlies its VDR antagonistic activity.

**ZK has a therapeutic effect for 1,25D3 intoxication**. To determine the impact of ZK on 1,25D3-induced calcium trafficking, we analyzed the intracellular calcium flux in IEC-18 cells pre-treated for 48 h with 100 nM 1,25D3 alone or in combination with 1 μM ZK. After $Ca^{2+}$ addition, intracellular $Ca^{2+}$ levels ($[Ca^{2+}]_i$) were 2.5-fold more increased in 1,25D3-treated cells than in vehicle-treated cells. In contrast, $[Ca^{2+}]_i$ were similar in 1,25D3 and ZK co-treated cells, and in vehicle-treated cells (Supplementary Fig. 4), demonstrating that ZK hampers 1,25D3-induced calcium flux.

These results prompted us to investigate the in vivo effects of ZK. In agreement with previous results[16], mice treated for 4 days with 1 μg/kg/day 1,25D3 were hypercalcemic. In contrast, serum calcium levels of mice co-treated with 1,25D3 and ZK for 4 days were similar to those of vehicle-treated mice (Fig. 5a). WBP4 was detected in cytosolic and nuclear extracts of duodenum and kidney of wild type mice (Supplementary Fig. 5a). In addition, WBP4 weakly interacted with VDR in kidney from wild-type mice (Supplementary Fig. 5b), and a 1,25D3 and ZK co-treatment enhanced this interaction (Fig. 5b). Moreover, VDR binding to vitamin D response elements (VDRE) located in the regulatory

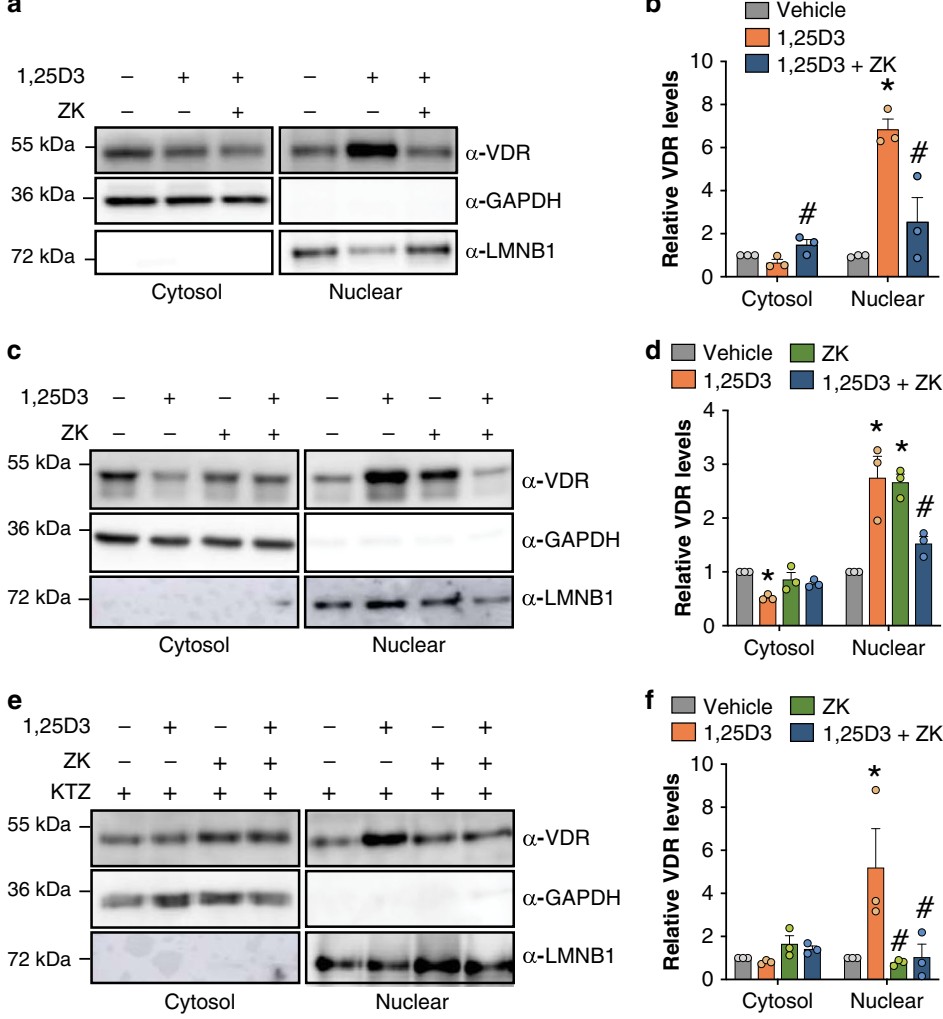

**Fig. 3 Ligand-dependent VDR subcellular localization.** Representative immunoblot (**a**) and quantification (**b**) of cytosolic and nuclear VDR in FB-789 cells treated for 1.5 h with vehicle, 10 nM 1,25D3, or 10 nM 1,25D3 and 100 nM ZK. GAPDH and LMNB1 were used as internal controls. $n = 3$ independent biological replicates/condition. Representative immunoblot (**c** and **e**) and quantification (**d** and **f**) of cytosolic and nuclear VDR in extracts of IEC-18 cells treated for 1.5 h with vehicle, 100 nM 1,25D3, 1 μM ZK, or 100 nM 1,25D3 and 1 μM ZK, pre-treated (**e** and **f**) or not (**c** and **d**) with 5 μM ketoconazole (KTZ) for 2 h. GAPDH and LMNB1 were used as internal controls. $n = 3$ independent biological replicates/condition. Unprocessed blots in Source Data. Data in (**b**), (**d**), and (**f**) are represented as mean + s.e.m. *$p < 0.05$ vs. vehicle, #$p < 0.05$ vs. 1,25D3; two-way ANOVA with Tukey post-hoc test. The exact significant $p$-values are provided in Supplementary Table 4.

regions of *Cyp24a1*, *Slc30a10*, *Slc37a2*, *Nkain1* and *Atp2b1*[28] was increased by 1,25D3 by at least 3-fold in mouse intestine, but not by a 1,25D3 and ZK co-treatment (Fig. 5c).

To gain insight into the mechanisms underlying ZK activities, genome-wide analysis of duodenum from mice treated for 6 h with vehicle, 1,25D3, or 1,25D3 and ZK was performed. The transcript levels of 3186 genes (1876 up and 1310 down) were modulated by at least 1.5-fold in the intestine of 1,25D3-treated mice compared to vehicle (Fig. 5d). Pathways analyses revealed that 1,25D3 induces the expression of genes involved in protein digestion, protein and mineral absorption, and cell adhesion, whereas it reduces that of genes contributing to CYP-induced metabolism, arachidonic acid, cholesterol and linoleic acid metabolism, and bile secretion (Supplementary Fig 6 and Source Data). In addition, the expression of >99% of the 1,25D3-regulated genes was normalized by a ZK-cotreatment (Fig. 5d), and only 14 genes were differentially expressed in the intestine of mice co-treated with 1,25D3 and ZK compared to vehicle (Source Data file Fig. 5d), none being a known VDR target gene

according to available databases[28]. To confirm that ZK efficiently blunts 1,25D3-induced gene expression, the transcript levels of the VDR target genes *Cyp24a1*, *Slc30a10*, *Slc37a2*, *Nkain1* and *Atp2b1* were determined in the duodenum of additional wild type mice. Their levels were induced by at least 3-fold by 1,25D3, but were similar in 1,25D3 and ZK co-treated mice and vehicle-treated mice (Fig. 5e). In addition, these transcripts were at least 5-fold higher in wild-type mice than in VDR-null mice when co-treated (Fig. 5e), and serum calcium levels remained physiological in co-treated wild-type mice whereas VDR-null mice were hypocalcemic (Fig. 5a). Thus, our results demonstrate that ZK does not fully abrogate VDR signaling in mice, but normalizes the expression of 1,25D3-induced target genes and hypercalcemia.

To determine the therapeutic potency of ZK after 1,25D3 intoxication, supra-physiological doses of 1,25D3 (1 μg/kg/day) were orally administered to mice for 2 days, followed by a 2-day treatment with 1,25D3 in the presence or absence of ZK (1 μg/kg/day) (Fig. 6a). Whereas mice treated with 1,25D3 alone were

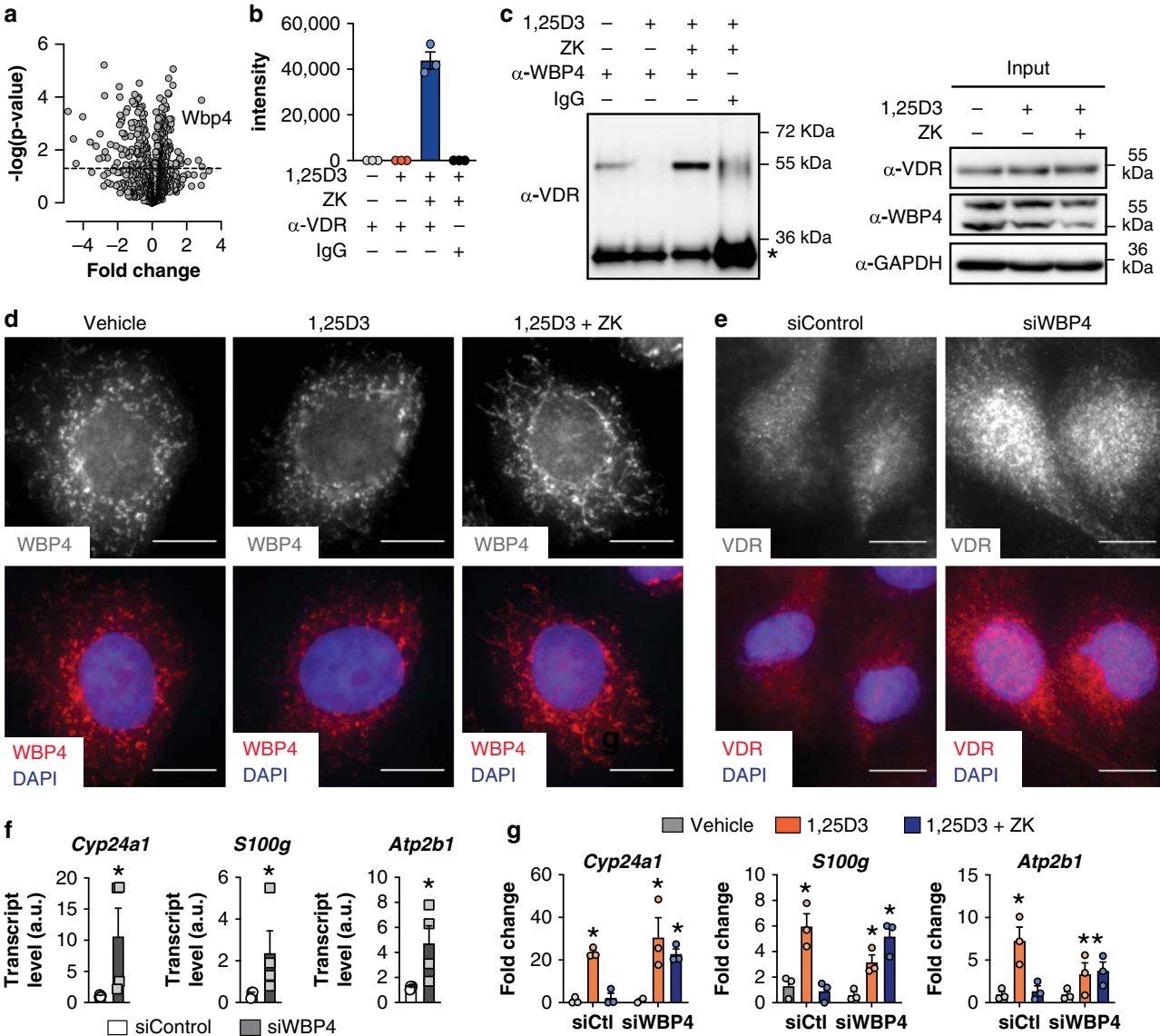

**Fig. 4 VDR interactants, intracellular localization, and transcriptional activities. a** Volcano plot representing peptides identified by mass-spectrometry of proteins bound to cytosolic VDR in cells co-treated with 100 nM 1,25D3 and 1 μM ZK relative to vehicle-treated cells. n = 3 technical replicates/condition. The dashed line represents a p-value of 0.05. Source Data are provided as a Source Data file. **b** Abundance of WBP4 peptides determined by mass-spectrometry of VDR-immunoprecipitated cytosolic extracts of IEC-18 cells treated for 1.5 h with vehicle, 100 nM 1,25D3 or 100 nM 1,25D3 and 1 μM ZK. Cytosolic extracts from IEC-18 cells treated for 1.5 h with 100 nM 1,25D3 and 1 μM ZK immunoprecipitated with IgG were used as a negative control. n = 3 technical replicates/condition. **c** Representative immunoblot of VDR in WBP4-immunoprecipitated cytosolic extracts of IEC-18 cells treated for 1.5 h with vehicle, 100 nM 1,25D3 or 100 nM 1,25D3 and 1 μM ZK. Cytosolic extracts from IEC-18 cells treated for 1.5 h with 100 nM 1,25D3 and 1 μM ZK immunoprecipitated with Rabbit IgG were used as a negative control (left panel). *Indicates antibody light chains. Representative immunoblot of VDR, WBP4, and GAPDH (used as loading control) in cytosolic extracts of IEC-18 cells treated for 1.5 h with vehicle, 100 nM 1,25D3 or 100 nM 1,25D3 and 1 μM ZK (Input, right panel). Unprocessed blots in Source Data. **d** Representative immunostaining of WBP4 in IEC-18 cells treated for 1.5 h with vehicle, 100 nM 1,25D3, or with 100 nM 1,25D3 and 1 μM ZK (top panels). Superposition with DAPI-stained nuclei (bottom panels). Scale bar: 10 μm. n = 2 independent biological replicates/condition. **e** Representative VDR immunostaining in hVDR-transfected IEC-18 cells silenced (siWBP4) or not (siControl) for WBP4 (top panels). Superposition with DAPI-stained nuclei (bottom panels). Scale bar: 10 μm. n = 3 independent biological replicates/condition. **f** Transcript levels of Cyp24a1, S100g and Atp2b1 in IEC-18 cells silenced (siWBP4) or not (siControl) for WBP4. *p < 0.05 vs. siControl, Student's t-test. n = 4 biological replicates/condition. **g** Transcript levels of Cyp24a1, S100g and Atp2b1 in IEC-18 cells transfected with WBP4 siRNA (siWBP4) and unrelated siRNA (siCtl) treated for 6 h with vehicle, 100 nM 1,25D3, or with 100 nM 1,25D3 and 1 μM ZK. n = 3 independent biological replicates/condition. *p < 0.05 vs. vehicle, one-way ANOVA with Tukey post-hoc test. Data in (**b**), (**f**) and (**g**) are represented as mean + s.e.m. The exact significant p-values are provided in Supplementary Table 4.

hypercalcemic with suppressed PTH levels, a 2-day treatment with ZK normalized serum calcium and PTH levels (Fig. 6b, c). Moreover, ZK blunted 1,25D3-induced Cyp24a1 transcript levels in the duodenum and kidney (Fig. 6d, e). Thus, ZK normalizes acute and chronic hypercalcemia induced by vitamin D₃ intoxication.

## Discussion

Various diseases are characterized by hypercalcemia secondary to high vitamin D₃ levels. In addition to limit calcium and vitamin D intake, current treatments are based on drugs lowering serum calcium levels. However, they do not target VDR, are poorly efficient, and induce major side effects.

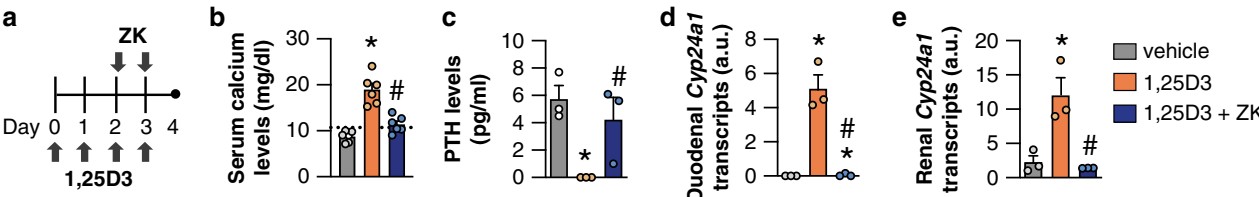

**Fig. 5 ZK activities in mice. a** Wild type mice were daily treated per os for 4 days with 1 μg/kg 1,25D3, 1 μg/kg 1,25D3 and 0.3 μg/kg ZK, 1 μg/kg 1,25D3 and 1 μg/kg ZK or vehicle, and serum calcium levels were determined 24 h after the last administration. The gray area represents the normocalcemic range. Age-matched VDR-null mice were used as hypocalcemic controls. $n = 3$ mice/condition. **b** Representative immunoblot of VDR in WBP4 immunoprecipitated (IP-αWBP4) or in total (input) extracts from kidney of wild type mice treated for 6 h with vehicle, 1 μg/kg 1,25D3, or 1 μg/kg 1,25D3 and 1 μg/kg ZK, and of VDR-null mice treated with 1 μg/kg 1,25D3 and 1 μg/kg ZK. Unprocessed blots in Source Data. **c** VDR ChIP-qPCR analysis of DNA segments encompassing vitamin D response elements of *Cyp24a1*, *Slc30a10*, *Slc37a2*, *Atp2b1*, and *Nkain1* from duodenum of wild type mice treated for 1.5 h with vehicle, 1 μg/kg 1,25D3, and 1 μg/kg 1,25D3 and 1 μg/kg ZK. $n = 3$ mice/condition. **d** Heatmap representing the mean-centered normalized transcript levels that were significantly different in the intestine of mice treated with 1,25D3 or with 1,25D3 and ZK, compared to vehicle. $n = 3$ mice/condition. **e** *Cyp24a1*, *Slc30a10*, *Slc37a2*, *Atp2b1* and *Nkain1* transcript levels in duodenum of wild type mice treated for 6 h with vehicle, 1 μg/kg 1,25D3, and 1 μg/kg 1,25D3 and 1 μg/kg ZK, and in VDR-null mice treated with 1 μg/kg 1,25D3 and 1 μg/kg ZK. $n = 4$ wild type mice/condition, except for *Atp2b1* ($n = 3$). $n = 3$ VDR-null mice/condition, except for *Atp2b1* ($n = 2$). Data in (**a**), (**c**), and (**e**) are represented as mean + s.e.m. *$p < 0.05$ vs. vehicle, #$p < 0.05$ vs. 1,25D3, \$$p < 0.05$ vs. 1,25D3 and ZK, one-way ANOVA with Tukey post-hoc test. The exact significant *p*-values are provided in Supplementary Table 4.

**Fig. 6 ZK therapeutic effect in mice. a** Schematic representation of 1,25D3 and ZK administration to mice. Serum calcium (**b**) and PTH (**c**) levels, and duodenal (**d**) and renal (**e**) relative *Cyp24a1* transcript levels, 24 h after the last administration of vehicle, 1 μg/kg 1,25D3, or 1 μg/kg 1,25D3 and 1 μg/kg ZK. Dashed line in (**b**) represents the maximal normo-calcemic concentration (10.7 mg/dL). **b** $n = 6$ mice/condition. **c**–**e** $n = 3$ mice/condition. Data are represented as mean + s.e.m. *$p < 0.05$ vs. vehicle, #$p < 0.05$ vs. 1,25D3, one-way ANOVA with Tukey post-hoc test. The exact significant *p*-values are provided in Supplementary Table 4.

ZK, a carboxylic ester analog of 1,25D3, was shown to antagonize 1,25D3-induced VDR activity in cultured cells, but the underlying molecular mechanisms were poorly characterized[10,11,13–15] and its in vivo activities unknown.

Here, we show that ZK normalizes the expression of 1,25D3-induced VDR target genes in rat intestinal cells and human fibroblasts. Unexpectedly, these genes were induced in various cell lines treated with ZK, even though, in agreement with previous results[14], ZK did not promote the agonistic conformation of VDR H12 in vitro. We demonstrate that CYP24A1-generated ZK derivatives have VDR agonistic activities, whereas ZK is a pure antagonist.

Interestingly, we show that 1,25D3-induced VDR nuclear translocation is impaired by ZK. We identified proteins associated with VDR in the cytosol, and unraveled WBP4, as a cytosolic VDR interactant. WBP4 was previously reported as a nuclear protein within the spliceosome complex[29], and its overexpression increased pre-mRNA splicing efficiency[26]. We demonstrate that the region of WBP4 encompassing two WW domains that are docking sites for proline-rich motifs[27] interacts with VDR, and ZK enhances the interaction. As ZK stabilizes less efficiently than 1,25D3 the VDR NLS encompassing a diproline motif (154-RPPVR-158), it is likely that ZK-bound VDR interacts with WBP4 through this region. Previous results showed that the affinity of WBP4 WW domains for splicing factors increased with the valency of proline-rich sequences[25]. However, as the concentration of the WBP4 polypeptides used in the microscale thermophoresis experiments to characterize WBP4/VDR interaction in the presence or absence of ZK did not allow to reach a saturation plateau, the binding mode between WBP4 and liganded- and unliganded-VDR remains to be determined.

Our results demonstrate that WBP4 is essential to maintain VDR in the cytosol and for ZK-induced VDR antagonistic activity. In contrast, it is dispensable for 1,25D3-induced VDR transcription and is not mandatory for RNA splicing, as the primers used to quantify various transcripts in WBP4-silenced cells spanned exon–exon junctions. Interestingly, we identified two major WBP4 isoforms that are differentially located in the cytosol and nucleus of various cell lines, in agreement with recent genome-wide studies suggesting that WBP4 is a multifunctional protein[30,31]. Whether the shorter one, which is more abundant in the cytosol, selectively interacts with VDR requires further investigation.

Importantly, we demonstrate that serum calcium levels of mice treated with 1,25D3 to induce hypercalcemia were within the physiological range when co-treated with ZK, and almost all intestinal genes modulated by 1,25D3 were unaffected. Moreover, a 2-day ZK treatment of 1,25D3-intoxicated mice normalized serum calcium and PTH levels. Thus, this compound, which selectively targets VDR, opens new therapeutic strategies.

Taken together, our results show that the vitamin D analog ZK impairs 1,25D3-induced VDR nuclear translocation by enhancing the interaction of the receptor with WBP4 in the cytosol, and prevents and/or normalizes 1,25D3-induced hypercalcemia in mice. Moreover, as ZK blunts 1,25D3-induced VDR target gene transcription in fibroblast from an IIH patient, but does not fully suppress VDR signaling in mice, this VDR antagonist represents a potent and safe therapeutic option for acute and chronic hypercalcemia secondary to hypervitaminosis D.

## Methods

**Chemical.** 1,25-Dihydroxyvitamin D3 (17936, Sigma Aldrich) and ZK168281 (ZK, gift from the Medicinal Chemistry Department of Schering AG) were dissolved in absolute ethanol at $10^{-2}$ M and stored at $-20\,°C$. Ketoconazole (KTZ, TOKU-E) was dissolved in DMSO at $10^{-2}$ M and stored at $-20\,°C$.

**Cell lines.** Mycoplasma-free IEC-18 rat intestinal epithelial cells (American Type Culture Collection) were grown in Dulbecco's modified Eagle's medium (DMEM) 4.5 g/L glucose supplemented with 5% fetal calf serum (FCS), 1 mM sodium pyruvate, 0.1 UI/mL insulin and 40 μg/mL gentamicin. The U2OS human bone osteosarcoma epithelial cells and the HeLa human cervical epithelial carcinoma cells were grown in DMEM 1 g/L glucose supplemented with 5% FCS and 40 μg/mL gentamicin. Cells at 80% confluency were grown for 48 h in medium in which FCS was charcoal-treated.

**Human fibroblasts.** The collection and use of the human fibroblasts was performed in adherence with the principles of the declaration of Helsinki. Written informed consent of a 55-year-old man with elevated urine calcium and 1,25D3 levels in spite of a markedly low parathormone concentration was obtained for collection of clinical and laboratory data, DNA collection, and use of skin biopsy derived fibroblast (authorization 941 from the Ethical Committee from Caen University Hospital). The 25-hydroxy-vitamin $D_3$/24,25-dihydroxy-vitamin $D_3$ blood ratio determined by liquid-chromatography tandem MS indicated a CYP24A1 deficiency[18]. Sanger sequencing of the *CYP24A1* coding sequence revealed c.62del/p.(Pro21Argfs*8) (rs744432244) and c.427_429del/p.(Glu143del) (rs777676129) mutations in a compound heterozygous state (reference transcript NM_000782.5). Three mm skin punches obtained from the inner left arm were dissociated and cultured in Gibco™ Amniomax™ Medium (ThermoFisher Scientific, Waltham, Massachusetts, USA) for 10 days. Cells were then grown in DMEM/Nutrient Mixture F-10 Ham (1:1) supplemented with 10% FCS and 40 μg/mL gentamicin. FB-789 normal human primary fibroblasts (IGBMC, France)[19] were maintained in Minimum Essential Medium (MEM) supplemented with 15% FCS, non-essential amino acid and antibiotics (40 μg/mL gentamicin, 100 UI/mL penicillin, and 100 μg/mL streptomycin).

**Gene silencing.** Small interference RNA (siRNA) transfection was performed following Dharmacon's protocol. Briefly, IEC-18 cells were seeded and grown overnight in serum-deprived growth medium to 80% confluency, transfected for 96 h with 1 μM siRNA targeting rat WBP4 (Accell pool ID: E-087189-00-0050) or non-targeting siRNA (ID: D-001910-10-50) in Accell delivery media, and treated as indicated.

**Mice.** C57BL/6 J mice (WT) and VDR-null (VDR$^{-/-}$)[32] mice on a similar genetic background were housed in a temperature- and light-controlled animal facility and fed ad libitum (Safe diets, D04, France). Ten-week-old male mice were administered per os with 100 μL of vehicle, 1,25D3 and/or ZK dissolved in oil (sunflower oil, Auchan). At the end of the treatment, blood was collected by inferior palpebral vein puncture, and mice were killed by cervical dislocation. Tissues were collected and immediately processed or frozen in liquid nitrogen. All animal experimental protocols were conducted in compliance with French and EU regulations on the use of laboratory animals for research, and approved by the IGBMC Ethical Committee and the French Ministry (#10047-2017052615101492 and #21776-2019082318288737).

**Serum calcium and PTH levels.** Blood was kept overnight at 4 °C and centrifuged at 400 g for 10 min at 4 °C. Serum calcium and PTH levels were determined using colorimetric assays (MAK022, Sigma Aldrich and Ab230931, Abcam, respectively) according to the supplier's protocol.

**RNA isolation and analysis.** Total RNA was isolated from mouse tissue and cells using NucleoSpin kit reagents (Macherey-Nagel GmbH & Co. KG) and TRI Reagent (Molecular Research Center, Inc.), respectively, according to the manufacturer's protocols. RNA was quantified by spectrophotometry (Nanodrop, Thermo Fisher), and cDNA prepared using 2 μg of total RNA, random hexamers and SuperScriptII reverse transcriptase (Thermo Fisher) following the manufacturer's instructions. Quantitative PCR reactions were performed using the Light Cycler 480 SYBR Green I Master X2 Kit (Roche) according to the supplier's protocol. Oligonucleotides are listed in Supplementary Table 3.

**Transcriptomic analysis.** cDNA libraries were generated from 600 ng of total RNA using the TruSeq Stranded mRNA LT Sample Preparation Kit (Illumina), according to the manufacturer's instructions, quantified and checked for quality using capillary electrophoresis. Fifty base pair single-read sequencing was performed on a Hiseq 4000 sequencer (Illumina) following the manufacturer's instructions. Image analysis and base calling were performed with RTA 2.7.7 and bcl2fastq 2.17.1.14 softwares. Adapter dimer reads were removed using Dimer-Remover. FastQC 0.11.2 was used to assess the quality of sequencing. Reads were mapped onto the mm10 assembly of the mouse genome using Tophat 2.1.1[33] and Bowtie2 2.3.4.3[34]. Only uniquely aligned reads were retained for further analyses. Quantification of gene expression was performed using HTSeq-0.11.0[35]. Read counts were normalized across libraries by the method of Anders et al.[36]. Comparisons of the transcripts with more than 100 raw reads were performed by the method of Love et al.[37] implemented in the DESeq2 Bioconductor library (DESeq2 v1.0.19). Resulting p-values were further adjusted for multiple testing. Genes were considered to be differentially expressed if the adjusted p-value was less than 0.05

and the |log2 Fold-change | > 0.58. Hierarchical clustering was performed using Cluster 3.0, and the heatmaps were visualized using the Java TreeView software. Gene Ontology annotation was performed using clusterProfiler[38].

**Protein isolation.** Harvested cultured cells and homogenized tissues (Precellys, Bertin instruments) were centrifuged at 400 g for 5 min. Cell pellets were resuspended in radioimmunoprecipitation assay lysis buffer (RIPA) [50 mM Tris pH 7.5, 1% Nonident P40, 0.5% sodium deoxycholate, 0.1% SDS, 150 mM NaCl, 5 mM EDTA, 1 mM PMSF, and phosphatase and protease inhibitor cocktails (PhosphoStop and Complete-Mini EDTA free; Roche)]. Lysates were cleared by centrifugation at 10,000 g for 10 min at 4 °C, and supernatant protein concentrations were determined using Bradford Reagent (ab119216, Abcam) according to the manufacturer's instructions. For subcellular fractionation, cell pellets were resuspended in 200 µL of 10 mM HEPES, 60 mM KCl, 1 mM EDTA, 0.075% (v/v) NP40, 1 mM DTT, and 1 mM PMSF (pH 7.6), and incubated on ice for 8 min. After centrifugation at 400 g for 5 min, supernatants were collected as cytosolic fractions. Pellets were washed in PBS, resuspended in 100 µL of 20 mM Tris HCl, 420 mM NaCl, 1.5 mM MgCl₂, 0.2 mM EDTA, 1 mM PMSF and 25% (v/v) glycerol (pH 8.0), and incubated for 10 min on ice. After centrifugation at 15,000 g for 10 min, supernatants were collected as the nuclear fractions.

**Immunoprecipitation.** One mg of total (mouse tissue) or cytosolic (IEC-18 cells) protein extracts obtained from fresh material was incubated overnight with 10 µL anti-VDR (D2K6W, Cell Signaling) or anti-WBP4 (Abcam) antibodies (Supplementary Table 3) in 500 µL immunoprecipitation buffer (50 mM Tris pH 7.5, 150 mM NaCl, 5% glycerol, 1% NP-40). Immunocomplexes recovered with G-coupled agarose beads (Sigma Aldrich) were washed and eluted in loading buffer (30 mM Tris HCl pH 6.8, 1% sodium dodecyl sulfate, 2.5% β-mercaptoethanol, 5% glycérol, and 0.001% bromophenol blue).

**Mass Spectrometry analysis.** Immunoprecipitated proteins were reduced, alkylated, and digested with trypsin at 37 °C overnight. Peptides were analyzed using an Ultimate 3000 nano-RSLC (Thermo Scientific) coupled in line with an Orbitrap ELITE (Thermo Scientific). Briefly, peptides were separated on a C18 nano-column with a linear gradient of acetonitrile and analyzed using the Top 20 collision-induced dissociation method. Data were processed by database searching against Rattus norvegicus Uniprot Proteome database (29944 sequences) with Maxquant 1.6.6.0 and Perseus 1.6.6.0. Precursor and fragment mass tolerance were set at 7 ppm and 0.6 Da, respectively. Trypsin was set as enzyme, and up to two missed cleavages were allowed. Oxidation and N-term acetylation were set as variable modification and carbamidomethylation as fixed modification. Proteins were identified with a minimum of two unique peptides and a false discovery rate <1%. Proteins identified in two out of three replicates were considered. p-values obtained by moderated t-test of median-normalized log2-data were transformed to a local false discovery rate of 0.05.

**SDS-PAGE analysis.** Equal amounts of proteins were resolved under denaturing conditions by electrophoresis in 8–12% SDS-containing polyacrylamide gels (SDS-PAGE) and transferred to nitrocellulose membranes (Trans-blot turbo transfer system, Bio-Rad) following the manufacturer's protocol. Membranes were incubated for 1 h at room temperature in 10 mM Tris pH 7.4, 0.05% Tween-20 (TBST) supplemented with 5% nonfat dry milk. Immunoblotting with antibodies (Supplementary Table 3) was performed overnight in TBST with 5% bovine serum albumin (BSA), and immunocomplexes were revealed upon a 1 h incubation with horseradish peroxidase (HRP)-conjugated antibodies directed against rabbit Ig (Jackson ImmunoResearch) with an enhanced chemiluminescence detection system (ECLplus, GE Healthcare) and an ImageQuant LAS 4000 biomolecular imager (GE Healthcare). For co-immunoprecipitation experiments, membranes were incubated with mouse anti-rabbit IgG (L27A9, Conformation Specific, Cell signaling) for 1 h at room temperature before addition of the secondary antibodies. Immunodetected proteins were quantified with FIJI/ ImageJ distribution Software[39]. The unprocessed scans of the immunoblots are available as a Source Data file.

**Chromatin immunoprecipitation.** Mouse duodenum was harvested and washed twice in ice-cold PBS, incubated in PBS containing 1% paraformaldehyde (PFA, Electron Microscopy Sciences) for 10 min, and quenched with 0.125 M glycine at room temperature for 5 min. Nuclei from cross-linked tissues were isolated in ChIP Buffer (ChIP-IT High Sensitivity, Active Motif) following the manufacturer's procedure and sonicated in 300–400 bp DNA fragments. Thirty µg of chromatin was immunoprecipitated with anti-VDR antibodies (D2K6W, Cell Signaling) and DNA was recovered following the manufacturer's directions (ChIP-IT High Sensitivity, Active Motif). qPCR reactions were performed using the Light Cycler 480 SYBR Green I Master X2 Kit (Roche) according to the supplier's protocol. Oligonucleotides are listed in Supplementary Table 3.

**Immunocytochemistry.** Cells grown in 8-well Lab-Tek II RS Glass Chamber Slides (Thermo Fisher) were washed twice with PBS, fixed in 4% PFA for 15 min at room temperature, washed in PBS and permeabilized in PBS 0.2% Triton X-100 for 5 min. After 1 h incubation in PBS-5% BSA, primary antibodies (Supplementary Table 3) were added overnight at 4 °C. After three washes in PBS, cells were incubated with fluorescent cyanine (Cy3)-conjugated anti-rabbit antibodies (Jackson ImmunoResearch Laboratories) for 1 h at room temperature and nuclei were stained with 4′,6-Diamidino-2-phenylindole (DAPI; Sigma Aldrich). Slides were examined under an epifluorescence microscope (Leica), and images were acquired with a 16-bit Cool Snap FX camera by the Leica Application Suite X (LAS X) software. Fluorescence intensity was determined using FIJI/ ImageJ distribution Software[39].

**Intracellular calcium measurement.** IEC-18 cells were grown in Ca²⁺-free medium [20 mM Hepes, 40% (w/v) glucose, 140 mM NaCl, 5 mM KCl, 1 mM MgCl₂] in the presence of 1 µM pluronic acid F-127 (Thermo Fisher) and 5 µM Indo-1 AM calcium probe (Thermo Fisher) for 30 min at 37 °C, washed two times with PBS, and incubated for 30 min at 37 °C in Ca²⁺-free medium. Ratiometric calcium imaging was performed using an inverted Leica SP8 UV/Visible Laser Confocal Microscope equipped with a Leica HC PL APO 63 × 1.4 N.A. oil immersion objective. Cells were subjected to a 355 nm UV laser. Two confocal images in the spectral range 400–440 nm and 470–540 nm were simultaneously recorded every 5 s using HyD detectors in the counting mode. Variation of the intracellular calcium concentration was determined by the intensity ratio between the two emission bands with the FIJI/ ImageJ distribution Software[39].

**Biochemistry.** DNA segments encoding His-tagged LBD of hVDR (amino acid 118–427) and of hRXRα (amino acid 223–462), and His-tagged hRXRαΔNTD (amino acid 130–462) were inserted into pET15b. The cDNA encoding His-tagged hVDR (amino acid 1–427) was cloned into pET28b. Recombinant proteins were produced in Escherichia coli BL21 DE3 after induction with 1 mM IPTG (OD600 ~ 0.7) at 23 °C for 4 h (hVDR full-length and hRXRαΔNTD) or at 18 °C overnight (LBDs of hVDR and of hRXRα). Soluble proteins were purified using affinity chromatography columns (HisTrap FF crude, 17-5286-01, GE) followed by size exclusion chromatography (HiLoad Superdex 200, 28-9893-35 GE) equilibrated in 20 mM Tris-HCl, pH 8.0, 250 mM NaCl, 5% glycerol, 2 mM CHAPS and 1 mM TCEP. Proteins were concentrated to 3–6 mg/mL with an Amicon Ultra 10 kDa MWCO and the purity and homogeneity were assessed by SDS- and Native-PAGE. For HDX-MS analysis, full-length hVDR and hRXRαΔNTD were mixed in stoichiometric amounts and purified by size exclusion chromatography (HiLoad Superdex 200, 28-9893-35, GE) equilibrated in 20 mM Tris pH 8.0, 75 mM NaCl, 75 mM KCl, 2 mM CHAPS, 5% Glycerol, 4 mM MgSO₄, 1 mM TCEP.

The cDNA encoding WBP4 amino acid 122-196 was inserted into pnEAtG[40] to allow the expression of a N-terminal GST-tagged protein. The recombinant protein was produced in Escherichia coli BL21 DE3 after induction at an OD600 of ~0.9 with 0.5 mM IPTG for 2.5 h at 20 °C. Soluble proteins were purified by glutathione sepharose (GE) chromatography, followed, after GST clivage by thrombin proteolysis, by size exclusion chromatography (HiLoad Superdex 200, 28-9893-35 GE) equilibrated in 20 mM Hepes pH 7.5, 250 mM NaCl, 5% glycerol, 2 mM CHAPS, and 1 mM TCEP. The recombinant proteins were concentrated to 2–5 mg/mL with an Amicon Ultra 3 kDa MWCO and their purity and homogeneity were assessed by SDS-PAGE.

**Surface plasmon resonance.** Measurements were performed with a Biacore T100 sensitivity enhanced T200 equipment (GE Healthcare) using CM5 series S sensor chip. hRXRα LBD monomers were immobilized on the chip surface in the range of 400–500 response unit using a standard amino-coupling protocol in 10 mM Na-acetate buffer pH 5.5. The running buffer was 50 mM Tris pH 7.5, 150 mM NaCl, 1 mM TCEP, 0.005% Tween 20; 1 M sodium chloride solution was used for regeneration. hVDR LBD (1–30 µM) was incubated with a 3-fold excess of ligand, and interactions between immobilized-hRXRα LBD were analyzed. The association and dissociation phase were 120 s. After subtracting the reference and buffer signal, the data were fit to a steady state binding model using the Biacore T200 Evaluation software (GE Healthcare).

**HDX-MS.** HDX-MS experiments were performed following the recommendations described by Masson et al.[41]. Full-length hVDR/hRXRααΔNTD heterodimers (10 µM) were incubated with a 10-fold excess of ligand or an equal volume of ethanol (vehicle) for 30 min at room temperature. Exchange reactions, performed with a CTC PAL sample handling robot (LEAP Technologies), were conducted by incubating 40 picomole of protein in 50 µl of D₂O buffer (50 mM Tris pH 8.0, 200 mM NaCl, 2% Glycerol, 1 mM TCEP) for 30, 60, 180, 600, or 1800 s. The exchange reaction was quenched by the addition of 50 µl 3 M Urea containing 0.1% TriFluoro Acetic acid followed by immediate injection into a Aspergillopepsin protease 2.1 × 20 mm column (Affipro) for protein digestion (2 min at 24 °C). Peptic digest was injected on a Waters nanoACQUITY UPLC System, where peptides were first desalted by trapping for 3 min on a VanGuard Pre-Column Acquity UPLC BEH C18 (1.7 µm; 2.1 × 5 mm), and eluted over 9 min with 5–40% (vol/vol) acetonitrile (containing 0.1% formic acid) gradient into a Waters Synapt G2-Si mass spectrometer. Peptide separation was conducted at 0.1 °C. All exchange reactions were performed in triplicates. As a reference, peptides from three non-

deuterated samples were identified using the ProteinLynx Global Server (PLGS Waters, UK). Peptides with an intensity over 5000, a mass error <5 ppm and present in at least two of the three data acquisitions were pooled and imported into the DynamX data analysis software (Waters, UK). After a first round of automated spectral processing using DynamX, each peptide was inspected manually for suitability for further analysis. For the full-length hVDR, a total of 179 peptides were identified with a 92.1% coverage and a mean redundancy of 3.99 per amino acid. For hRXRαΔNTD, a total of 127 peptides were identified with an 88.7% coverage and a mean redundancy of 3.98 per amino acid. MEMHDX software was used to identify statistically significant changes of deuterium uptake that occurred upon compound binding[42]. The mean deuteration level per amino acid was calculated using Matlab (Mathworks) and subsequently mapped onto the crystal structures with PyMOL (Schrodinger LLC). The cryoEM structure[19] was used for the representation of the hVDR/hRXRα LBD heterodimer. HDX-MS data have been deposited to the ProteomeXchange Consortium via the PRIDE partner repository[43].

**Native PAGE analysis**. Recombinant hVDR full-length was mixed with a 5-fold molar excess of hWBP4 polypeptide (amino acid 122-196) in the presence of two equivalents of ZK. Then, 5 μg of proteins were loaded on a 4–15% gradient GE Healthcare PhastGel™ with Native PhastGel Buffer Strips (GE) following the manufacturer's protocol and revealed by Coomassie Brilliant Blue (CBB) staining.

**Microscale thermophoresis**. Measurements were performed with a Monolith NT.115 instrument (NanoTemper Technologies GmbH). Recombinant hVDR full-length and hWBP4 polypeptide (amino acid 122-196) were prepared in 20 mM Hepes pH 7.5, 200 mM NaCl, 2% Glycerol, 1 mM TCEP, and 0.05% Tween 20. The full-length hVDR was labeled with Red-NHS second generation labeling kit (NanoTemper) and incubated with ZK. The labeling procedure and the subsequent removal of free dye were performed within 1 h. The solution of unlabeled WBP4 was serially diluted from a concentration of 100 μM down to 1 nM in the presence of 10 nM labeled receptor. Measurements were made with standard glass capillaries (Nanotemper) at 25 °C, at 40% LED excitation and 80% MST power. Isotherms were averaged over three consecutive measurements and fitted according to the law of mass action with a 1:1 stoichiometry to yield the apparent Kd, by using the NanoTemper Analysis 2.2.4 software.

**Data analysis**. No inclusion/exclusion criteria, and no method of randomization were used in this study. No blinding was used for animal studies. Data are represented as mean + standard error of mean (s.e.m.). The samples followed a normal distribution, and the variances were similar. Statistical comparisons of data between two groups were made by a Student's $t$-test and those between three and more by one-way ANOVA followed by a post-hoc analysis (Tukey's test). Data were considered to be statistically significant if $p < 0.05$, and are indicated by *, $ or # in the figures (GraphPad Software Prism 8). The exact significant $p$-values are provided in Supplementary Table 4.

**Reporting summary**. Further information on research design is available in the Nature Research Reporting Summary linked to this article.

## Data availability
RNA sequencing raw data are available in GEO database (GSE141985). Mass spectrometry raw data of IEC-18 cells are available as a Source Data file Fig. 3a. HDX-MS raw data are available via ProteomeXchange with identifier PXD019810. All relevant data are available from the authors. Source data are provided with this paper.

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

## Acknowledgements

We thank the IGBMC animal house facility, molecular biology, cell culture and antibody services, proteomic and imaging platforms, and GenomEast, a member of the 'France Génomique' consortium (ANR-10-INBS-0009), as well as R. Lutzing, C. Kostmann, C. Peluso-Iltis, and M. Aubert for excellent technical assistance, and P. Antony for helpful discussions. We acknowledge the Medicinal Chemistry Department of Schering AG (Bayer) for the ZK compound. This work was supported by INSERM, CNRS, Unistra and IGBMC, and by an INSERM young researcher fellowship and French state funds from Agence Nationale de la Recherche ANR-VARaD to G.L. and the grant ANR-10-LABX-0030-INRT, a French State fund managed by the Agence Nationale de la Recherche under the frame program Investissements d'Avenir ANR-10-IDEX-0002-02. A.Y.B. is a fellow of the AstraZeneca R&D post-doctoral program, and A.I.R. an IGBMC International PhD Programme fellow supported by LabEx INRT funds.

## Author contributions

D.R., N.R., G.L., and D.M. conceived the study and analyzed the results. D.R. performed in vitro and in vivo experiments. A.Y.B. performed and analyzed HDX-MS. S.C. produced proteins used for HDX-MS and performed SPR. A.I.R. analyzed the sequencing data. E.G. helped to design and analyze fluorescent microscopy experiments. A.M. provided FB-CYP. A.L. discussed the results. N.R., G.L., and D.M. wrote the manuscript with input from all the authors. All authors edited the manuscript and approved the final manuscript.

## Competing interests

The authors declare no competing interests.
