## [Peer Review File · Nature Communications]

Reviewers' comments:

Reviewer #1 (Remarks to the Author):

Major comments:

1. The statement "its mechanism of action were unknown" is not fully correct, since initial papers already described a mechanism.

2. Peräkylä et al (Chemistry & Biology, Vol. 11, 1147–1156) reported species-specific differences in the mechanisms of antagonism. Therefore, it is important to confirm the gene regulatory findings also in a human experimental setting and not only in rodents.

3. The transcriptome-wide results presented in Fig. 4D were only analyzed and described superficially. This should be extended and build into the model understanding the actions of ZK.

Minor comments:

1. All gene name abbreviations should be in italic, this applies also to tables and figures. In general, nomenclature and abbreviation style should be harmonized (e.g. *Wbp4* and *WBP4*).

2. It may be more helpful for the reader, if there is a consistent name for 1,25D3, i.e. better avoid the term "calcitriol".

3. The company "Schering" is already since more than 10 year part of BAYER.

Reviewer #2 (Remarks to the Author):

The interaction of VDR and Wbp4/FBP21 in the article by Rovito et al. is investigated by IP experiments performed with extracts of IEC-18 cells. While the authors performed a differential centrifugation step to enrich the cytoplasm, I would be very cautious in stating that most of the protein is cytoplasmic in this fraction. I think the authors should provide a full list of proteins found in the IP. For example, what about known binding partners of WBP4, such as Brr2 helicase or Prp8 ?

The primarily cytoplasmic localization of WBP4 is in contrast to what has been seen in most other cells. Moreover, the authors of this paper suggest that WBP4 acts as an anchor that keeps the receptor in the cytoplasm and prevents its nuclear translocation. This is extremely difficult to conceptualize. Wbp4 is one of the spliceosomal B complex proteins (Betram et al., Cell, 2017) and as a regulator of Brr2 helicase (Henning et al., NAR, 2017) it seems to be an important protein for ensuring splicing fidelity. Why would such a protein act as a cytoplasmic anchor? While proteins can have multiple functions, the question arises whether there is something particular about the IEC-18 cells and whether the non-nuclear localization of FBP21 affects the splicing pattern of these cells ? Also, the antibody used for immunofluorescence is recommended for western blotting but not necessarily for immunofluorescence. While the siRNA experiment shown in the extended Figure 3 lends support to their hypothesis about cytoplasmic localization of WBP4, the observation could still be an off-target effect. Certainly, more controls are required, especially since the WBP4 localization is so unusual. A different antibody should be used and the interaction should also be confirmed in vitro. WBP4 can be well expressed in E. coli and thus the interaction be mapped by HDX, ITC, Biacore or else.

On an additional note: according to the human protein atlas, WBP4 is expressed ubiquitously and staining exists also for the small intestine. The protein is reported to stain as a nuclear protein in this case. Since IEC-18 cells are a model of the small intestine where the VDR protein is expressed, I wonder how this discrepancy can be explained? In conclusion, I think more direct evidence for a direct interaction and cytoplasmic localization of the complex has to be shown. In case the conclusion turns out to be correct, the finding needs to be conceptualized with regard to the finding in other cells and the anticipated function of the VDR/WBP4 axis outside of the nucleus.

Reviewer #3 (Remarks to the Author):

In this manuscript, Dr Rovito and co-workers presented a novel Vitamin D receptor (VDR) antagonist, ZK, as a therapeutic option for calcitriol-induced hypercalcemia. The authors used a combination of techniques, such as HDXMS, SPR, proteomics, transcriptomic analysis and Immunocytochemistry to

reveal the molecular mechanism of how ZK regulate calcitriol-induced gene expression. This manuscript is well written and constructed and can be accepted for publication after the following comments being addressed.

1. Pg 4, line 2: “ Moreover, deuterium exchange rates of hVDR amino acids 342-389 and of hRXRa amino acids 426-433, forming the dimerization interface, were similar upon ZK and 1,25D3 binding.....”

1) The authors only inserted figure for hRXRa 426-433, but not for hVDR 342-389, Why?

2) From HDXMS data, I cannot tell if these regions are forming dimer interface. If the conclusion is from literature, please cite it here.

3) hVDR and hRXRa are similar in size, why their binding interface size are quite different (47 vs 7)?

4) Extended data Fig2 d

The figure ligand doesn't match with line marker.

2. Pg4, line5: “...ligand binding domain....”

Where is ligand binding domain, it is unclear in the text and figure. Please clarify.

3. Pag4, line 9-14.

The author compared HDX results of hVDR between ZK-bind and CaCalcitriol-bind states. However, the figures are put in two different figures. It is better to move extended data Fig 2b and sc to Figure 2a and combine to one figure.

Reviewer #4 (Remarks to the Author):

In their manuscript Rovito et al show the effects of the VDR antagonist ZK168281 (ZK) in normalizing serum calcium levels induced by calcitriol in intoxicated mice and calcitriol-induced VDR signaling fibroblasts in IIH patient.

About the in vivo data, I do not believe that the model is appropriate. Induction of hypercalcemia with very high dose of calcitriol causes severe hypoparathyroidism (data are lacking). The use of a VDR Receptor antagonist in this condition is probably mediated by low PTH levels, and its effects on parathyroid glands should be further elucidated.

Reviewers' comments:

Reviewer #1 (Remarks to the Author):

We thank the reviewer for her/his valuable comments.

Major comments:

1. The statement "its mechanism of action were unknown" is not fully correct, since initial papers already described a mechanism.

We acknowledge that previous studies reported *in vitro*, *in cellulo* and *in silico* mechanisms of action of ZK. We summarize the state of the art **page 2 line 23 to page 3 lines 2** of our revised version. However, *in vivo* ZK activities remained to be determined.

2. Peräkylä *et al* (*Chemistry & Biology*, Vol. 11, 1147–1156) reported species-specific differences in the mechanisms of antagonism. Therefore, it is important to confirm the gene regulatory findings also in a human experimental setting and not only in rodents.

Peräkylä *et al* reported that the VDR antagonistic activity of TEI947 is species-specific, whereas that of ZK was similar in rat and human cells¹.

In our manuscript, we showed that ZK antagonizes 1,25D3-induced *CYP24A1* transcript levels in rat, mouse and human cells (Fig. 1 and Fig. 4), and that it induces specific conformational changes in recombinant human VDR different from those observed for 1,25D3 (Fig. 2). In addition, we revealed that the subcellular localization of rat VDR was altered by ZK (Fig. 2). To extend our data, we analyzed VDR subcellular localization in human-derived fibroblasts (FB-789 cells). Our new results demonstrate that ZK impairs 1,25D3-induced VDR nuclear localization in both rat and human cells (**Extended data Fig. 3a,b**). Moreover, we show that WBP4 is located in the cytosol of FB-789 cells, and that a 1,25D3 and ZK co-treatment increases VDR and WBP4 interaction in human cells (**Extended data Fig. 3e,f**). Thus, our results demonstrate that the mechanism underlying ZK antagonistic activity is not species-specific. These additional experiments are described **page 5 lines 4-10** and **page 5 line 23-25**.

3. The transcriptome-wide results presented in Fig. 4D were only analyzed and described superficially. This should be extended and build into the model understanding the actions of ZK.

To further analyze the transcriptomic data, we compared the intestinal transcript levels in vehicle, 1,25D3-treated and in 1,25D3 and ZK co-treated mice, and determined the enriched pathways by KEGG annotation. The main text (**page 7 line 22 to page 8 line 6**), **Fig. 4d**, material and methods (**page 11 line 21**) and **Extended data Fig. 4c** were amended.

Minor comments:

1.1 All gene name abbreviations should be in italic, this applies also to tales and figures. In general, nomenclature and abbreviation style should be harmonized (e.g. *Wbp4* and *WBP4*).

As suggested, we harmonized the abbreviations. Please note that whereas human genes are in italicized upper cases (e.g. *WBP4*), only the first letter is in upper-case for rodent genes (e.g. *Wbp4*). In contrast, proteins are in upper cases for both species.

1.2 I may be more helpful for the reader, if there is a consistent name for 1,25D3, i.e. better avoid the term "calcitriol".

As suggested by the reviewer, we now substituted calcitriol by 1,25D3 in the text and figures.

1.3 The company "Schering" is already since more than 10 year part of BAYER.

We corrected the statement **page 19 line 6** of the revised manuscript.

Reviewer #2 (Remarks to the Author):

We thank the reviewer for her/his relevant remarks.

1. The interaction of VDR and Wbp4/FBP21 in the article by Rovito et al. is investigated by IP experiments performed with extracts of IEC-18 cells. While the authors performed a differential centrifugation step to enrich the cytoplasm, I would be very cautious in stating that most of the protein is cytoplasmic in this fraction. I think the authors should provide a full list of proteins found in the IP. For example, what about known binding partners of WBP4, such as Brr2 helicase or Prp8 ?

Western blot analysis of the subcellular fractions revealed that GAPDH and LMNB1 were selectively immunodetected in the cytosolic and nuclear extracts, respectively (Fig. 2f-g, Extended data Fig. 3a,e,i,n). As GAPDH and LMNB1 expression are restricted to the cytosol and the nucleus, respectively, we conclude that the supernatant obtained after the centrifugation step is enriched in cytosolic proteins. In addition, cellular component terms annotation by Gene Ontology (GO) showed that VDR co-immunoprecipitated proteins enriched in 1,25D3 and ZK co-treated cells are mainly localized in the cytosol (**Extended data Table 2**). Importantly, even though WBP4 peptides were selectively detected in the cytosolic extract from 1,25D3 and ZK co-treated cells (Fig. 3b), none of its previously identified nuclear partners [e.g. BRR2 and PRP8²] was detected. We now added these informations in the main text **page 5 line 17-19 and line 22-23**.

2. The primarily cytoplasmic localization of WBP4 is in contrast to what has been seen in most other cells. Moreover, the authors of this paper suggest that WBP4 acts as an anchor that keeps the receptor in the cytoplasm and prevents its nuclear translocation. This is extremely difficult to conceptualize. Wbp4 is one of the spliceosomal B complex proteins (Betram et al., Cell, 2017) and as a regulator of Brr2 helicase (Henning et al., NAR, 2017) it seems to be an important protein for ensuring splicing fidelity. Why would such a protein act as a cytoplasmic anchor? While proteins can have multiple functions, the question arises whether there is something particular about the IEC-18 cells and whether the non-nuclear localization of FBP21 affects the splicing pattern of these cells ? Also, the antibody used for immunofluorescence is recommended for western blotting but not necessarily for immunofluorescence. While the siRNA experiment shown in the extended Figure 3 lends support to their hypothesis about cytoplasmic localization of WBP4, the observation could still be an off-target effect. Certainly, more controls are required, especially since the WBP4 localization is so unusual. A different antibody should be used and the interaction should also be confirmed in vitro. WBP4 can be well expressed in E. coli and thus the interaction be mapped by HDX, ITC, Biacore or else. On an additional note: according to the human protein atlas, WBP4 is expressed ubiquitously and staining exists also for the small intestine. The protein is reported to stain as a nuclear protein in this case. Since IEC-18 cells are a model of the small intestine where the VDR protein is expressed, I wonder how this discrepancy can be explained? In conclusion, I think more direct evidence for a direct interaction and cytoplasmic localization of the complex has to be shown. In case the conclusion turns out to be correct, the finding needs to be conceptualized with regard to the finding in other cells and the anticipated function of the VDR/WBP4 axis outside of the nucleus. *In conclusion, I think more direct evidence for a direct interaction and cytoplasmic localization of the complex has to be shown. In*

case the conclusion turns out to be correct, the finding needs to be conceptualized with regard to the finding in other cells and the anticipated function of the VDR/WBP4 axis outside of the nucleus.

Recent genome wide studies indicate that WBP4 might be a multifunctional protein involved in protein transport^{3,4}. In addition, according to the human protein atlas, WBP4 has a cytoplasmic and nuclear localization in renal cells. Moreover, even though it is reported that 75% of WBP4 is nuclear in the small intestine, a cytosolic staining is clearly observed. As these results were obtained using an antibody targeting WBP4 amino acids 15 - 113 (Abcam ab272629), we now performed WBP4 immunoblotting of nuclear and cytosolic extracts from mouse duodenum and kidney, as well as from HeLa, U2OS, FB789 and IEC-18 cells, using this antibody (**Extended data Fig. 3e,i and Extended data Fig. 4a**). Our results show that WBP4 has a nuclear and cytosolic localization in two mouse tissues and cells from various species. Note that only the WBP4 Abcam ab272629 antibody detects 2 forms, in agreement with the data sheet, that their relative amount is cell- and localization-dependent, and that these 2 bands were not detected after WBP4-silencing in IEC-18 cells (**Extended data Fig. 3k**).

In addition, we demonstrate that ZK promotes the interaction between VDR and WBP4 in cytosolic extracts from IEC-18 and in FB-789 cells, using the two Abcam WBP4 polyclonal antibodies, targeting either aa 1-50 or aa 15 -113 (**Fig. 3f and Figure 1**).

Figure 1. (a) Representative VDR immunoblot of WBP4-immunoprecipitated cytosolic extracts from IEC-18 cells treated for 1.5 h with vehicle, 100 nM 1,25D3 (calcitriol), or 100 nM 1,25D3 and 1 μ M ZK. Cytosolic extracts from IEC-18 cells treated for 1.5 h with 100 nM 1,25D3 and 1 μ M ZK and immunoprecipitated with Rabbit IgG were used as a negative control. WBP4 polyclonal antibodies targeting either aa 1-50 or aa 15 -113 were used.

Moreover, to demonstrate that VDR interacts with WW domains of WBP4, we performed native gel experiments and microscale thermophoresis analyses using recombinant VDR and the WBP4 polypeptide 122-196 (**Extended data Fig. 3g,h**). Our results show that their interaction is enhanced by the ZK analog.

Whereas WBP4 overexpression has been shown to enhance splicing^{2,5}, our data show that WBP4-silencing has no general effect on mRNA splicing. Indeed, RT-qPCR analyses performed with primers spanning exon-exon junctions amplified similar cDNAs in WT and WBP4-silenced cells, and that the levels of several of them were increased in the latter.

We addressed these points in the main text **page 6 line 3-22 and page 7 line 1-5, and in Extended data Fig 3e-i,k**. In addition, the material and methods section was amended (**page 16 line 2-10, page 18 line 1-16**).

Note that, whether WBP4 acts as a cytosolic anchor for other proteins, alone or within a larger complex, remains to be determined, but is out of scope of this study.

Reviewer #3 (Remarks to the Author):

In this manuscript, Dr Rovito and co-workers presented a novel Vitamin D receptor (VDR) antagonist, ZK, as a therapeutic option for calcitriol-induced hypercalcemia. The authors used a combination of techniques, such as HDXMS, SPR, proteomics, transcriptomic analysis and Immunocytochemistry to reveal the molecular mechanism of how ZK regulate calcitriol-induced gene expression. This manuscript is well written and constructed and can be accepted for publication after the following comments being addressed.

We are pleased that the reviewer appreciated our manuscript and thank for his/her valuable comments.

1. Pg 4, line 2: “ Moreover, deuterium exchange rates of hVDR amino acids 342-389 and of hRXRa amino acids 426-433, forming the dimerization interface, were similar upon ZK and 1,25D3 binding.....”

The authors only inserted figure for hRxRa 426-433, but not for hVDR 342-389, Why? From HDXMS data, I cannot tell if these regions are forming dimer interface. If the conclusion is from literature, please cite it here. hVDR and hRXRa are similar in size, why their binding interface size are quite different (47 vs 7)?

We previously identified the residues of the heterodimer interface of RXR [i.e H7(353-361), H9(395-410), H10(417-435)] and of VDR [i.e. L8-9(340-344), H9(350-368) and H10(377-395)] ⁶. The amino acid range indicated in the first version of the manuscript corresponded to the RXR peptides identified by HDX in these helices, rather than the heterodimerization interface itself.

As suggested, we now show in the revised version the position of peptides localized on H10 of RXR α and of VDR that form the major dimer interface, and the HDX of amino acid 388 – 393 of VDR H10 (see **Extended data Fig. 2a-f**). We modified the text accordingly (**page 4 line 10**).

2. Extended data Fig2 d

The figure ligand doesn't match with line marker.

The Figure has been modified in order to fit with the legend.

3. Pg4, line5: “...ligand binding domain....”

Where is ligand binding domain, it is unclear in the text and figure. Please clarify.

To clarify this point, we annotated the helices forming the ligand binding domain in the main text **page 4 line 14** and in **Figure legend 2a**.

4. Pag4, line 9-14.

The author compared HDX results of hVDR between ZK-bind and Calcitriol-bind states. However, the figures are put in two different figures. It is better to move extended data Fig 2b and sc to Figure 2a and combine to one figure.

As suggested by the reviewer, we inserted the panel from **Extended data Fig 2b** into **Fig. 2a**. The Figure legend and the main text were modified accordingly.

Reviewer #4 (Remarks to the Author):

In their manuscript Rovito et al show the effects of the VDR antagonist ZK168281 (ZK) in normalizing serum calcium levels induced by calcitriol in intoxicated mice and calcitriol-induced VDR signaling fibroblasts in IIH patient.

About the in vivo data, I do not believe that the model is appropriate. Induction of hypercalcemia with very high dose of calcitriol causes severe hypoparathyroidism (data are lacking). The use of a VDR Receptor antagonist in this condition is probably mediated by low PTH levels, and its effects on parathyroid glands should be further elucidated.

We thank the reviewer for these comments and the opportunity to clarify this point.

Calcium homeostasis is controlled by a network involving the parathyroid glands (PTG), intestine, kidney and bones. PTG, by modulating the secretion and/or production of PTH, plays a central role in sensing low serum calcium levels. PTH enhances the hydroxylation of vitamin D into 1,25D3 that in turn, via VDR, induces calcium absorption in intestine, calcium reabsorption in kidney, and under certain circumstances promotes calcium mobilization from bones. Importantly, high 1,25D3 levels increase VDR-mediated intestinal calcium absorption leading to hypercalcemia, and shut down PTH production in PTG via calcium-dependent but VDR-independent pathways ⁷. A section providing additional background information on calcium homeostasis has been added in the main text of the revised version **page 2 line 12-18**.

To study 1,25D3 intoxication, wild type mice were administered for 5 days with a regimen of 1,25D3 known to induce hypercalcemia ⁸. As expected, we now show that 1,25D3-induced hypercalcemia results in low/suppressed PTH (**Fig. 4f,g**). Thus, such a regimen represents a suitable model to study the therapeutic potency of ZK for 1,25D3 intoxication. Moreover, we show that ZK normalizes serum calcium levels, as well as PTH levels (**Fig. 4g**).

These additional data are described **page 8 line 18-19**.

References

- 1 Perakyla, M., Molnar, F. & Carlberg, C. A structural basis for the species-specific antagonism of 26,23-lactones on vitamin D signaling. *Chem Biol* **11**, 1147-1156, doi:10.1016/j.chembiol.2004.05.023 (2004).
- 2 Henning, L. M. *et al.* A new role for FBP21 as regulator of Brr2 helicase activity. *Nucleic Acids Res* **45**, 7922-7937, doi:10.1093/nar/gkx535 (2017).
- 3 Chapple, C. E. *et al.* Extreme multifunctional proteins identified from a human protein interaction network. *Nat Commun* **6**, 7412, doi:10.1038/ncomms8412 (2015).
- 4 Bassaganyas, L. *et al.* New factors for protein transport identified by a genome-wide CRISPRi screen in mammalian cells. *J Cell Biol* **218**, 3861-3879, doi:10.1083/jcb.201902028 (2019).
- 5 Huang, X. *et al.* Structure and function of the two tandem WW domains of the pre-mRNA splicing factor FBP21 (formin-binding protein 21). *J Biol Chem* **284**, 25375-25387, doi:10.1074/jbc.M109.024828 (2009).
- 6 Orlov, I., Rochel, N., Moras, D. & Klaholz, B. P. Structure of the full human RXR/VDR nuclear receptor heterodimer complex with its DR3 target DNA. *Embo J* **31**, 291-300, doi:10.1038/emboj.2011.445 (2012).
- 7 Meir, T. *et al.* Deletion of the vitamin D receptor specifically in the parathyroid demonstrates a limited role for the receptor in parathyroid physiology. *Am J Physiol Renal Physiol* **297**, F1192-1198, doi:10.1152/ajprenal.00360.2009 (2009).
- 8 Laverny, G. *et al.* Synthesis and anti-inflammatory properties of 1 α ,25-dihydroxy-16-ene-20-cyclopropyl-24-oxo-vitamin D₃, a hypocalcemic, stable metabolite of 1 α ,25-dihydroxy-16-ene-20-cyclopropyl-vitamin D₃. *J Med Chem* **52**, 2204-2213, doi:10.1021/jm801365a (2009).

REVIEWER COMMENTS

Reviewer #1 (Remarks to the Author):

All my comments were addressed to my satisfaction. Only in Fig. 1D the "Calcitriol" had been forgotten to be replaced by "1,25D3"

Reviewer #2 (Remarks to the Author):

The authors have adequately addressed my concerns, but I still have one comment that needs to be resolved. When the authors now show the thermophoreses results they conclude that the affinity is 2 fold higher for the ZK compound. I'am not convinced by the data, since the saturation level in Ext. data 3h is also different. I think, this is for sure not the reason to explain the action of the compound. Rather, it looks to me that the stoichiometry is distinct for VitD3 and the derivative. FBP21 has two WW domains, and they prefer to bind as tandem domains (Klippel et al., 2011). For the ZK derivative, presumably only one WW domain is bound to one receptor molecule (see observed 1:1 stoichiometry), while it seems to be two receptors bound to one tandem construct in case that VitD3 is the saturating ligand ? Is it possible that such a change in stoichiometry contributes to the observed differences in transport ? Maybe the authors could share their reasoning with me or ideally implement it in the discussion of their results.

Reviewer #3 (Remarks to the Author):

The Authors have addressed the reviewer's concerns and have made corrections in the text. The manuscript can be accepted for publication.

Reviewer #4 (Remarks to the Author):

Authors answered to my comments

Reviewers' comments

Reviewer #1 (Remarks to the Author):

All my comments were addressed to my satisfaction. Only in Fig. 1D the "Calcitriol" had been forgotten to be replaced by "1,25D3"

We are pleased that the reviewer was satisfied by the revised version of our manuscript.

One "Calcitriol" was indeed not replaced in one of the figures. The change has now been made in Figure 2D.

Reviewer #2 (Remarks to the Author):

The authors have adequately addressed my concerns, but I still have one comment that needs to be resolved. When the authors now show the thermophoreses results they conclude that the affinity is 2 fold higher for the ZK compound. I'am not convinced by the data, since the saturation level in Ext. data 3h is also different. I think, this is for sure not the reason to explain the action of the compound. Rather, it looks to me that the stoichiometry is distinct for VitD3 and the derivative. FBP21 has two WW domains, and they prefer to bind as tandem domains (Klippel et al., 2011). For the ZK derivative, presumably only one WW domain is bound to one receptor molecule (see observed 1:1 stoichiometry), while it seems to be two receptors bound to one tandem construct in case that VitD3 is the saturating ligand ? Is it possible that such a change in stoichiometry contributes to the observed differences in transport ? Maybe the authors could share their reasoning with me or ideally implement it in the discussion of their results.

We are happy to see that we addressed all reviewer's concerns, and to have the opportunity to share our reasoning on his/her final comment, and to implement it in the discussion of our results.

In our original manuscript we had demonstrated by co-immunoprecipitation that WBP4 interacts with VDR in intestinal rat cells and in mouse kidney (Figure 3c, Figure 4b). To determine whether VDR and WBP4 directly interact, the reviewer proposed to investigate the interaction *in vitro*. We thus determined whether recombinant full length VDR formed a complex with the tandem WW domains of WBP4 (amino acid 122 – 196) by native gel electrophoresis. Our results clearly show that the two polypeptides directly interact (Extended Figure 2g).

In addition, we performed Microscale thermophoresis experiment to further analyse this interaction, by titrating unlabeled WBP4 polypeptide (amino 122 -196) into a fixed concentration of fluorescently labeled monomeric recombinant VDR in the absence of ligand (apo) or in the presence of two equivalents of ZK. Isotherms averaged over three consecutive measurements were fitted according to the law of mass action in a 1:1 stoichiometry to yield the apparent K_d . Our data show that WBP4 tandem WW domains bind directly to monomeric apo and ZK-liganded VDR (Extended Figure 2h), and indicate that the affinity is increased by about 2-fold in the presence of ZK. Note that cooperative binding of the tandem WW domains of WBP4, as well as the multivalency of the proline rich ligands, have been shown to increase the affinity of WBP4 to splicing factors (Klippel et al., JBC 2011). However, as the concentrations used in our study did not allow to reach a saturation plateau, the valency of the interaction between tandem WW domains and monomeric VDR could not be determined. Deciphering the exact binding mode of WBP4 to apo or ZK-liganded VDR, and whether it influences VDR transport, will require additional biophysical and structural analysis, and is out of scope of this study.

To clarify these points, the text (page 6 line 5-13) was changed accordingly. Note that Extended Figure 3h was mislabeled, as 1,25D3 should have been apo. We corrected the figure, legend and M&M accordingly.

Reviewer #3 (Remarks to the Author):

The Authors have addressed the reviewer's concerns and have made corrections in the text. The manuscript can be accepted for publication.

We are pleased that the reviewer found our manuscript ready for publication.

Reviewer #4 (Remarks to the Author):

Authors answered to my comments

We are happy that we could answer all his/her comments, and further improve our manuscript.

REVIEWERS' COMMENTS

Reviewer #2 (Remarks to the Author):

The authors have addressed my final concerns.